
# Simulation and detection efficiency analysis for polar mesospheric clouds measurements using a spaceborne wide field of view ultraviolet imager

Ke Ren[2], Haiyang Gao[1,2], Shuqi Niu[2], Shaoyang Sun[2], Leilei Kou[1,2], Yanqing Xie[3], LiGuo Zhang[3], and Lingbing Bu[1,2]

[1]Key Laboratory for Aerosol-Cloud-Precipitation of China Meteorological Administration, Nanjing University of Information Science and Technology, Nanjing, 210044, China,
[2]School of Atmospheric Physics, Nanjing University of Information Science and Technology, Nanjing, 210044, China
[3]Shanghai Academy of Spaceflight Technology, Shanghai, 201109, China

*Correspondence to*: Haiyang Gao (gaohy@nuist.edu.cn)

**Abstract.**

The variation trends and characteristics of polar mesospheric clouds (PMCs) are important for studying the evolution of atmospheric systems and understanding various atmospheric dynamic processes. Through observation and analysis of PMCs, we can gain a comprehensive understanding of the mechanisms driving atmospheric processes, providing a scientific basis and support for addressing climate change. Ultraviolet (UV) imaging technology, adopted by the Cloud Imaging and Particle Size (CIPS) instrument onboard the Aeronomy of Ice in the Mesosphere (AIM) satellite, has significantly advanced the research on PMCs. Due to the retirement of the AIM satellite, there is currently no concrete plan for next-generation instruments based on the CIPS model, resulting in a discontinuity in the observation data sequence.

In this study, we propose a compact and cost-effective wide field-of-view ultraviolet imager (WFUI) that can be integrated into various satellite platforms for future PMCs observation missions. A forward model was built to evaluate the detection capability and efficiency of the WFUI. The fusion of CIPS and solar occultation for ice experiment (SOFIE) data was employed to reconstruct a three-dimensional PMCs scene as the input background. Based on the scattering and extinction characteristics of ice particles and atmospheric molecules, the radiative transfer was calculated using the solar radiation path through the atmosphere and PMCs. The optical system and satellite platform parameters of the WFUI were selected according to the CIPS, enabling the calculation of the number of photons received by the WFUI. The actual detection signal is then simulated by photoelectric conversion, and the PMCs information can be obtained by removing detector noise. Subsequently, a comparison with the input background field was conducted to compute and analyze the detection efficiency. Additionally, a sensitivity analysis of the instrument and platform parameters was conducted.

Simulations were performed for both individual orbits and for the entire PMCs seasons. The research results demonstrate that the WFUI performs well in PMCs detection and has high detection efficiency. Statistical analysis of the detection efficiency using data from 2008 to 2012 revealed an exponential relationship between the ice water content (IWC) of PMCs and detection efficiency. During the initial and final durations of the PMCs season, when the IWC was relatively low, the detection efficiency





remained limited. However, as the season progressed and the IWC increased, the detection efficiency significantly improved. We note that regions at lower latitudes exhibited a lower IWC and, consequently, lower detection efficiency. In contrast,

regions at higher latitudes, with a greater IWC, demonstrated better detection efficiency. Additionally, the sensitivity analysis results suggest that increasing the satellite orbit altitude and expanding the field of view (FOV) of the WFUI both contribute to improving the detection efficiency.

## 1 Introduction

Polar mesosphere clouds (PMCs), also referred to as noctilucent clouds (NLCs) as observed from the ground, are the highest

ice clouds in the Earth's atmosphere, boasting an average thickness of 2–5 km. PMCs generally form at approximately 83 km in high latitudes and are highly sensitive to the ambient atmosphere. Thus, the long-term trend in PMCs variation is considered an indicator of long-term changes of temperature and water vapor content in the Earth's atmosphere (Deland et al., 2015, 2019; Hervig et al., 2001; Hervig, Berger, et al., 2016; Thomas and Olivero, 1989). Prior research has indicated that the frequency and observed brightness of PMCs in the mid-latitudinal areas have increased over the past half-century. (Miao et al., 2022;

Kaifler et al., 2018; Tylor et al., 2017). PMCs are extremely sensitive to changes in the atmospheric temperature and respond to planetary, tidal, and gravity waves in the upper atmosphere (Gao et al., 2018; Liu et al., 2015; Stevens et al., 2010). The variation trends and characteristics of PMCs are important for studying the evolution of atmospheric systems and understanding various atmospheric dynamic processes.

Satellite-borne instrumental observations provide global coverage. Generally, occultation, nadir, and limb-viewing modes have

their own advantages and are adopted for different purposes. The solar occultation for ice experiment (SOFIE) onboard the aeronomy of ice in the mesosphere (AIM) satellite utilizes the occultation mode to observe the extinction properties of ice particles with 16 bands and inverts the mass density, radius, and other parameter profiles of the PMCs (Gordley et al., 2009). The detection sensitivity is excellent, particularly for small ice particles, and the vertical spatial resolution is as high as 0.2 km. However, the number of observation samples was limited; only two observations could be obtained at sunrise and sunset during

each orbit. In comparison, the limb viewing mode obtains PMCs profiles by scattering solar ultraviolet (UV) light using ice particles. SBUV/NOAA, Osiris/Odin, Sciamachy/Envisat and AHI/ Himawari-8 can be used to obtain the global distribution of PMCs using onion-peeling, tomography, and other inversion technologies (Deland et al., 2006; Broman et al., 2019; Savigny et al., 2004; Tsuda et al., 2021; Tsuda et al., 2018). Through these data, the understanding of the formation, evolution, and response of PMCs to climatic and background atmospheric conditions has been significantly enhanced.

The cloud imaging and particle size (CIPS) device onboard the AIM satellite is an instrument that adopts the nadir mode and uses UV imaging technology to observe PMCs (McClintock et al., 2009). The CIPS obtains a cloud map with a large horizontal range by recording the scattering of light through the atmosphere and ice particles (Rusch et al., 2009). It can provide panoramic cloud images, temperature, water vapor content, atmospheric dust density, and other data at a latitude of 45–85º daily in the PMCs season. For the first time, near-ultraviolet albedo data from PMCs have shown a panoramic cloud picture covering the





entire polar region. The commendable reliability of the data products has been widely recognized (Russell et al., 2009; Lumpe et al., 2013). Small-scale structures within various cloud layers can be observed using CIPS images with a high spatial resolution (Chandran et al., 2010, 2012; Gao et al., 2018). This further enhances our understanding of the impacts of small-scale gravity waves on PMCs. Owing to its large coverage, it has also improved the understanding of PMCs for large-scale dynamic processes, such as tidal waves, planetary waves, and even microphysical features (France et al., 2018; Liu et al., 2016;

Rusch et al., 2017). Nonetheless, the small sampling volume corresponding to the field of view of each pixel/bin of the CIPS resulted in relatively weakly scattered echo signals. Therefore, it is not sensitive to clouds with low ice water content (IWC) and smaller particles (Bailey et al., 2015). This causes significant inconsistencies in distinguishing the onset of the PMCs seasons, especially when compared with observations such as SOFIE and SBUV (Bardeen et al., 2010, Benze et al., 2009, 2011). Thus, it is necessary to systematically evaluate the detection efficiency of UV imaging technologies, such as CIPS, to

observe PMCs. However, CIPS is the only spaceborne instrument that adopts the nadir mode; therefore, the lack of other comparisons of the same type of data also makes it difficult to assess the detection efficiency.

AIM has ended its service, but the next generation of instruments based on UV imaging technology is not currently planned. This is not conducive to further research on PMCs. To address this issue, we propose a wide field-of-view ultraviolet imager (WFUI) designed to be compact, cost-effective, and integrated into various satellite platforms for future PMCs observation

missions. The main objective of this study was to construct a set of forward models for the WFUI. Section 2 describes the simulation method and its details. A 3D PMCs model was established as the detection target in Section 3 based on both CIPS and SOFIE data. Section 4 presents the simulation results and further discussion, and a concluding summary is provided in Section 5.

## 2 Simulation method

### 2.1 Observation geometry and principles

Similar to the CIPS, the WFUI was designed as a nadir camera to image sunlight scattering signals by ice particles of PMCs in the 265 nm UV band. Figure 1a demonstrates the reason for using the 265 nm UV band for WFUI. The designated band of UV radiation from the sun in space first reaches the PMCs and is scattered by the ice particles of the PMCs with a radius of 5–100 nm. The scattered light within the FOV of the WFUI was received and imaged. Large amounts of stratospheric ozone

strongly absorb solar UV radiation as it passes through the high-altitude layer. This prevents UV radiation from below altitudes of 40 km, which is scattered by the ground, lower atmosphere, and cloud/aerosol layer, from being uploaded, ensuring that it does not interfere with the detection signal at the PMCs altitude. However, the scattering of UV radiation by atmospheric molecules from 40 to 80 km causes weak interference in the detection of PMCs. The calculation results obtained from the Line-By-Line Radiative Transfer Model (LBLRTM) software, shown in Fig. 1b, indicate that the atmospheric transmittance

of solar radiation with wavelengths of 265 nm above 70 km was close to 1. This implies that the radiation can almost reach the PMCs without significant loss. Conversely, the curve in Fig. 1c demonstrates strong absorption of the atmosphere below



70 km in this band. This absorption is primarily attributed to the distribution of ozone in the stratosphere at altitudes ranging from 20 to 40 km, with a peak concentration occurring at altitudes ranging from 20 to 25 km(the distribution of ozone transmission with altitude is shown in Fig. 1d).

The WFUI, designed to be mounted on a sun-synchronous orbiting satellite, captures images at regular intervals (tens of seconds). Sequential images with overlapping areas can be synthesized to cover the entire orbit using multiple photographs. The camera utilizes an array of CCD (Charge-Coupled Device) sensors and employs coating technology to enhance the quantum efficiency of the sensor in the UV band. As depicted in Fig. 1a, because the WFUI was mounted on the satellite platform, the line-of-sight direction had an inclination angle ranging from 30 °–50 ° with the satellite orbit. The projection area

in the PMCs altitude layer resembles a sector, resulting in substantial variation in the projection area corresponding to each pixel of the CCD sensor. An identical light-sensitive unit corresponds to a smaller area in proximity to the imager and a significantly larger area further away from the imager. The specific calculation method is described in the following sections.

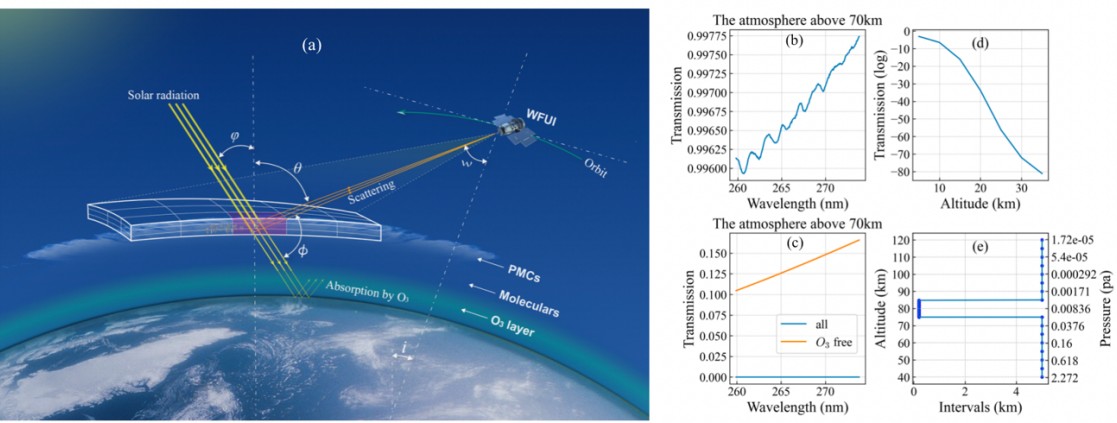

**Figure 1: Principle of WFUI for PMCs observation. (a) Observation geometry (the figure was created by ourself). (b)–(c)**
**Atmospheric transmittances above and below 70 km respectively for the wavelength of 265 nm. (d) the distribution of ozone**
**transmission with altitude. (e) the vertical distribution of atmospheric levels with corresponding heights and approximated**
**pressures.**

## 2.2 Forward model for simulation

The forward model is a comprehensive simulation of the instrument. It not only provides theoretical support for the design of
the instrument, but also provides necessary prior information and forward operators for subsequent data inversion. This study constructed a forward model consisting of seven sub-modules: 3D reconstructed PMCs datasets, ice particle scattering and extinction properties, atmospheric radiative transfer calculations, WFUI optical system and satellite platform model, detector signal calculation, and parameter sensitivity analysis. Figure 2 illustrates the principle and logical structure of the seven submodules of the forward model. By utilizing seven years of CIPS horizontal cloud images and SOFIE vertical profile data
for reconstruction and fusion, a three-dimensional PMCs dataset was established as the detection target of the forward model. The optical properties, such as the phase function and scattering cross-section of the ice particles and atmospheric molecules, were calculated. The observation model was integrated into the atmospheric radiative transfer calculation, and the number of



photons received by the WFUI was computed using the parameters of the WFUI optical system and satellite platform as the inputs. A CCD array detector model was constructed to simulate the actual signal detected through photoelectric conversion,

and the signal-to-noise ratio was obtained. The effective signals were extracted from the simulated detection results and compared and verified with the detection target to calculate and analyze the detection efficiency. A sensitivity analysis of multiple input parameters evaluated their impact on the detection performance. The key computational processes and key information in each module are listed in Sections 2.3–2.6.

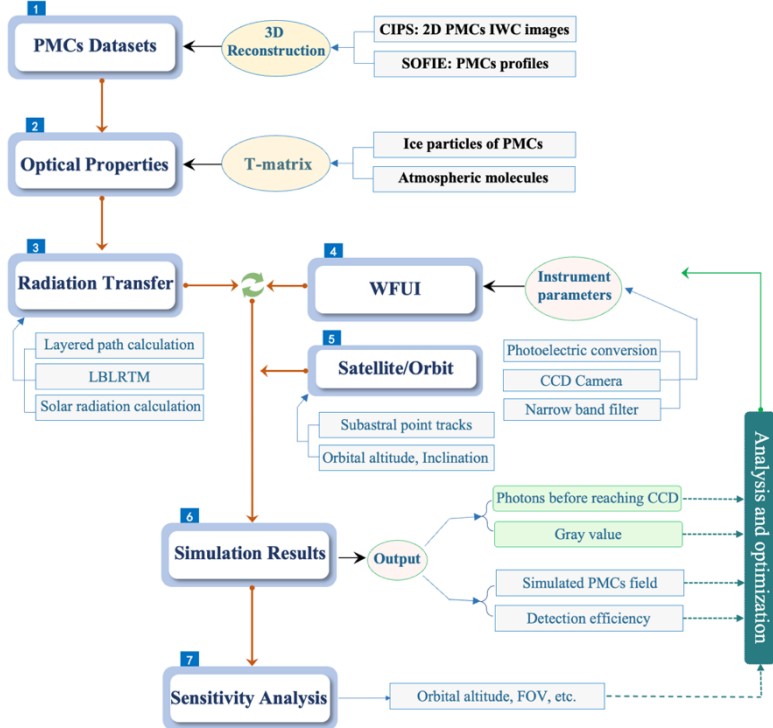

**Figure 2: The overall framework and principles of the forward model, and the logical structures among the seven sub-modules**

### 2.3 Optical properties

The PMCs are composed of ice particles ranging from 5 to 100 nm in size. Owing to the limited number density of ice particles ($\sim 10^3$ km$^{-3}$), the scattering effect outweighs the extinction absorption effect. Physical quantities, such as the scattering phase function, scattering cross-section, and extinction cross-section, were introduced to describe the scattering intensity. These

parameters are primarily influenced by the particle size, shape, and wavelength of the incident light. It is widely acknowledged that ice particles in PMCs are nonspherical in shape. In computational analyses, these non-spherical particles are treated as randomly oriented ellipsoids (Baumgarten et al., 2002). The axial ratio ($A_R$) represents the ratio of the horizontal axis to the rotation axis of an ellipsoid and effectively characterizes the degree of particle nonsphericity. The $A_R$ of ice particles in PMCs lies in the range of approximately 2 to 5 or 0.2 to 0.5. Using ground-based and satellite observations in 2007, the $A_R$ of ice



particles in PMCs most closely approximates 0.5 or 2, and the scattering characteristics of ice particles are essentially similar

for both $A_R$ values of 0.5 and 2 (Rapp et al., 2007).

In this study, the T-matrix was utilized to calculate the scattering properties of nonspherical particles (Xie et al., 2020). This

method is applicable to a wide range of uniformly symmetric particles and is one of the most powerful, widely used methods

for conducting rigorous calculations of light scattering from resonant non-spherical particles. The scattering cross-section,

extinction cross-section, and scattering phase function at a wavelength of 265 nm with particle radii ranging from 5 to 100 nm

and $A_R$ of 0.5, 2, and 3 were calculated by the T-matrix. In Fig. 3, for $A_R$ = 0.5 and 2, the scattering and extinction cross-

sections of ice particles are essentially the same, while they are slightly smaller for $A_R$ = 3. For particles with a radius greater

than 40 nm, smaller wavelengths resulted in decreased scattering and extinction cross-sections. For ice particles, the scattering

and extinction cross-sections are essentially identical, indicating that extinction is predominantly caused by scattering.

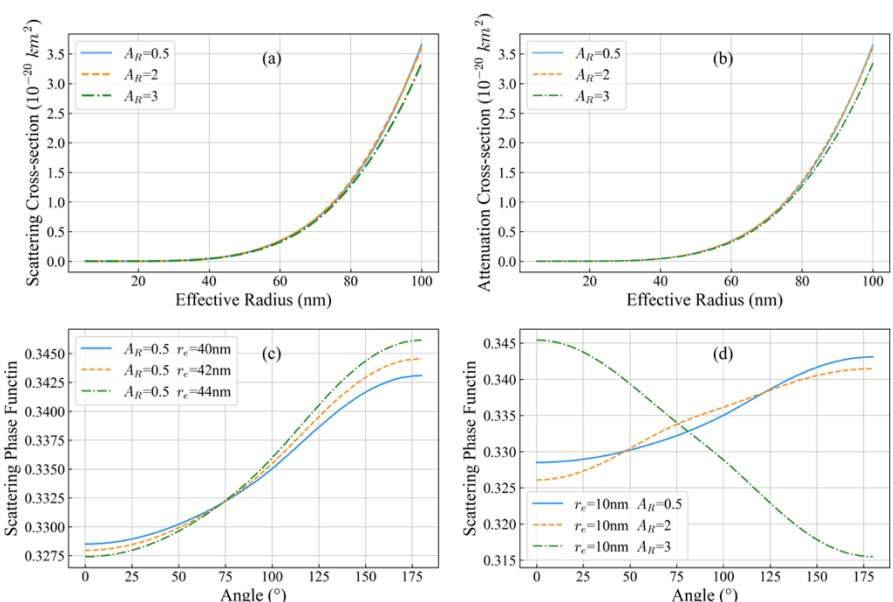


**Figure 3: Panels (a) and (b) shows the scattering and extinction cross-section regarding particle effective radius ($r_e$) respectively at different axis ratio ($A_R$ = 0.5, 2, 3). Panel (c) denotes the scattering phase function at WL = 265 nm for different $r_e$, while Panel (d) denotes the scattering phase function at $r_e$=10 nm for different $A_R$.**

The extinction and scattering of solar radiation by atmospheric molecules also influence the detection of PMCs signals. The

optical properties of atmospheric molecules can be calculated by the following equations:

$$P = \frac{\rho R T_a}{M} \tag{1}$$

$$N_a = 3000000 \frac{P}{K T_a} \tag{2}$$

$$m = \left( 6432.8 + \frac{2949810}{146 - \lambda_0^{-2}} + \frac{25540}{41 - \lambda_0^{-2}} \right) \div 10^8 + 1 \tag{3}$$



$$Q_a = \frac{8}{3.14} \times \frac{8\pi^4 r^6}{\lambda_0^4} \left(\frac{m^2 - 1}{m^2 + 2}\right)^2 (1 + cos^2\alpha) \qquad (4)$$

$$\sigma_a = \frac{128\pi^5 r^6}{3\lambda_0^4} \left(\frac{m^2 - 1}{m^2 + 2}\right)^2 \qquad (5)$$

where $P$, $\rho$, $R$, $T_a$ denote the atmospheric pressure, atmospheric density, proportionality constant, and atmospheric temperature, respectively; $M$, $N_a$, $K$, and $m$ are the atmospheric molar mass, number density of atmospheric molecules, Boltzmann's constant, and complex refractive index of atmospheric molecules, respectively; $\lambda_0$ represents the center wavelength of the WFUI; $Q_a$ and $\sigma_a$ are the scattering phase function and scattering cross-section of the atmospheric molecules, respectively; and $r$ is the average radius of the atmospheric molecules. The atmospheric density and temperature data were calculated using the NRLMSISE-00 Atmosphere Model. The calculation results were compiled into a database for the forward model.

## 2.4 Solar radiation

Solar radiation was crucial for our simulation because the WFUI signal relies on the scattering of solar radiation by ice particles. Under the condition of the Earth-Sun average distance, the solar radiation energy received per unit area perpendicular to sunlight at the upper boundary of the Earth's atmosphere is referred to as the solar constant. The wavelength range of the solar radiation was mainly distributed between 0.15 and 4 μm. For the energy distribution in a specific band, solar radiation can be calculated using the radiative transfer model libRadtran (Zhang et al., 2019). The WFUI was designed as an ultraviolet camera. Generally, the distribution of solar radiation in the ultraviolet range follows Planck's equations. During the calculation, we selected the wavelength bands near the center wavelength of the WFUI to compute the quantity of solar radiation. Subsequently, the radiation intensity within the spectral range covered by the WFUI was obtained by coupling with the instrument parameters. The solar radiation, $S$, for specific bands at the altitude of the PMCs (approximately 83 km) can be calculated as follows:

$$S = S_0 \times \sum_{\lambda=\lambda_0-\Delta\lambda}^{\lambda=\lambda_0+\Delta\lambda} (B_\lambda \times t_\lambda) = S_0 \times \sum_{\lambda=\lambda_0-\Delta\lambda}^{\lambda=\lambda_0+\Delta\lambda} \left( \frac{c_1}{\pi\lambda^5(e^{\frac{c_2}{\lambda T_s}} - 1)} \times \frac{t_{fmax}}{1 + \left[\frac{2(\lambda - \lambda_0)}{FWHM} + \frac{\lambda}{FWHM}\frac{\theta_0^2}{n_e^2}\right]^2} \right) \qquad (6)$$

where $S_0$ is the solar constant, $B_\lambda$ is the Planck function, $t_\lambda$ represents Lens transmittance which is a function of $\lambda$. $T_s$ is the temperature of the solar photosphere, which is approximately 6,000 K, $c_1$, $c_2$ is the constant, $t_{fmax}$ is the peak transmittance, FWHM is the full width at half maximum of WFUI, $\lambda_0$ is the central wavelength of the WFUI, and $\lambda$ represents the incident wavelength, with its value ranging within $\lambda_0$ plus or minus FWHM. The individual photon energy is $E_0 = hc/\lambda$, where $h = 6.62 \times 10^{-34} J \cdot s$ and $c = 3 \times 10^8 m/s$. Taking 1 nm as the interval, the number of photons ($N_0$) per second per unit area received at the altitude of the PMCs (approximately 83 km) within a specific band was calculated.



## 2.5 Radiation transfer calculation

When sunlight is transmitted through atmospheric media, if its optical thickness is small, single scattering can be employed to calculate the scattered light intensity and polarization characteristics. The optical thickness of PMCs is usually 0–3, which is consistent with the approximation of single cattering calculations. Due to the presence of PMCs within the altitude range of

75 km to 85 km, the altitude range of 75–85 km is meticulously segmented into 50 layers, each with intervals of 0.2 km. The altitude range of 40–75 km was divided into seven layers at 5 km intervals; and 85–120 km was divided into seven layers at 5 km intervals, resulting in a total of 64 layers(Fig. 1e).This approach to altitude layer segmentation not only maintains the accuracy of the calculation results but also effectively improves computational efficiency. As solar light passes through each layer, some photons are scattered or absorbed, whereas the remaining photons pass through the next layer. The remaining

photons then undergo scattering as they enter subsequent layers. A small number of photons are scattered in the direction of the satellite-borne detector; after multiple interactions with both atmospheric and PMC scattering and absorption, they are received by the detector lens.

When the WFUI observes PMCs, the spacecraft view angle, $\omega$, is defined as the angle between the line-of-sight (LOS) vector and the spacecraft nadir. The parameters of the WFUI were based on those of the AIM satellites, with $\omega$ set at 30 °. The most

important quantities for describing the scattering geometry, and hence the relevant CIPS radiative transfer problem, are the solar zenith angle, $\varphi$, view angle, $\theta$, and scattering angle, $\phi$. The satellite operates in a polar orbit 600 km above Earth. Based on the radius of Earth and the altitude of PMCs, the angle between the satellite to the center of Earth and PMCs to the center of Earth can be calculated to be approximately 2.7 °, resulting in a value of $\theta$ at 32.7 °. The solar zenith angle was constant throughout the season, ranging from the point of the descending node (approximately 20 °) to the transiting endpoint of the

ascending portion of the orbit (approximately 105 °). By utilizing CIPS data, the solar zenith angle can be obtained for different positions, and the scattering angle can then be calculated as $180° - \theta - \varphi$.

Each position corresponds to an IWC and $\phi$. Both scattering and extinction coefficients can be calculated from the optical properties (see Section 2.3). The radiative transfer characteristics of the entire atmosphere from 40 to 120 km can be obtained through the path integral of sunlight through the atmosphere and PMCs. Consequently, the number of photons, $N_\mathrm{p}$, received

per second per unit area at different locations can be determined:

$$N_p = N_0 \times (\beta_{sa} + \beta_{si}) \exp\left(-2\int_{h_1}^{h_2} (\beta_{ea} + \beta_{ei})\,dh\right) \qquad (7)$$

$$\beta_{si} = \int_{r=5}^{r=40} \sigma_i(r) \times Q_i(r,\phi) \times N_i(r,h_c,IWC) \qquad (8)$$

$$\beta_{sa} = \sigma_a \times Q_a(\phi) \times N_a(h_c,Lat) \qquad (9)$$

where $Q_i$ and $\sigma_i$ denote the scattering phase function and scattering cross-section of ice particles, respectively; $Q_a$ and $\sigma_a$

represent the scattering phase function and scattering cross-section of atmospheric molecules, respectively; $N_i$ corresponds to the scaled spectral distribution related to different heights and IWC, whereas $N_a$ indicates the concentration of molecules



related to different altitudes and latitudes; and $N_0$ represents the number of photons received per second per unit area at the altitude of the PMCs (approximately 83 km) (see Section 2.4 for the calculation procedure). The scattering coefficient was computed as the product of the scattering cross-section, molecular number concentration, and scattering phase function. As

both ice particles and atmospheric molecules exhibit weak absorption at 265 nm, the extinction coefficient can be substituted with the scattering coefficient.

**2.6 Image Calibration**

The longitude and latitude of the ground track of the AIM satellite can be calculated according to the longitude and latitude information from CIPS data. We assumed that the satellite carrying the WFUI operates in the same orbit and captures images

at a fixed time interval, $T_M$. The satellite moves around Earth at different operating speeds depending on its orbital altitude. The operating speed, $V$, is determined by $V = \sqrt{u/(R_0 + H)}$, where $u$ is the Keplerian constant, with a value of 398,610 km³/s², and $R_0$ is the radius of Earth. When the orbital altitude of the satellite was 600 km, its operating speed was calculated to be 7.56 km/s. After the simulated satellite travelled a fixed distance ($V \times T_M$) along the ground track, the WFUI captured an image. The exposure time for the CCD camera to capture an image was 0.75 s, during which the satellite covered a distance

of 5.67 km along its orbit, spanning an area corresponding to multiple pixels. This discrepancy is known as the phase-shift error. Data corresponding to multiple pixels were averaged to reduce the impact of the phase-shift error.

Owing to the non-uniformity of the light source, response difference of the photosensitive unit, and other factors, the CCD camera may capture an image with an uneven gray value for a target with a uniform gray value. The center of the image exhibited the highest gray value, which decreased as the image moved toward the edge. Therefore, it is necessary to perform

flat-field processing on the grey values obtained from the simulated WFUI. After normalizing the image center to 1, a pixel-by-pixel correction factor should be provided that can be applied to each image during the calibration process. For the CCD camera carried by the WFUI, the correction factor follows as a cosine function, with the center of the image assigned a value of 1, and the pixels located on the farthest edges of the circle having a correction factor of cos (90 °-$\omega$). However, more accurate flat-field coefficients should be obtained by experimental calibration after instrument development is complete.

**2.7 Optoelectronic conversion of signals**

After entering the optical system, the radiative photons undergo a photoelectric conversion process in the CCD sensor, transforming them into electrical signals. The number of electrons obtained per pixel is denoted by $N_{ele}$, *which* comprises two components: $N_{sca}$ represents the number of electrons obtained through the photoelectric conversion of solar radiation photons by scattering and $N_{noise}$ represents the number of electrons converted by various types of noise in the imager.

Subsequently, the electrons are converted into a digital signal by an analog-to-digital converter in accordance with the conversion factor, also known as the gray value, $U$. The gray value serves as a parameter for measuring the brightness of each





pixel in a gray image. In our model, CCD cameras have advanced to produce 16-bit gray-value images, offering a gray value ranging from 0 to $2^{16}$, with a richer color expression. The formula for calculating the grey value is as follows:

$$U = \frac{N_{ele}}{C_{ADU}} = \frac{N_{sca} + N_{noise}}{C_{ADU}} \tag{10}$$


$$N_{sca} = \eta \times T_e \times R_{at} \times N_p \times W \times H \tag{11}$$

$$N_{noise} = \sqrt{\sigma_s^2 + \sigma_D^2 + \sigma_R^2} \tag{12}$$

where $C_{ADU}$ denotes the analog-to-digital conversion coefficient; $\eta$ denotes the quantum efficiency of the CCD camera, estimated to be approximately 80 %, indicating that approximately 80 % of the photons can be converted into electrons following the Poisson distribution; $T_e$ is the exposure time of the CCD camera; $R_{at}$ denotes the photon receiving efficiency,

given by $R_{at} = \frac{\pi(D_c/2)^2}{4\pi(H-h_0)^2}$, where $D_c$ represents the diameter of the lens of the WFUI and $h_0$ is the altitude of the PMCs (approximately 83 km); and $N_p$ refers to the number of photons received per second per unit area by the WFUI (the calculation process is detailed in Section 2.5).

Here, $P_W$ and $P_H$ represent the spatial resolutions of the WFUI: $P_W$ denotes the length covered by each pixel along the satellite orbit, while $P_H$ is the length covered by each pixel across the satellite orbit. The CCD camera carried by the WFUI

had a resolution of 1,360 × 1,360 pixels. To expedite image processing and enhance the signal-to-noise ratio, adjacent 4 × 8 pixels (32 pixels) were merged, resulting in a final image size of 170 × 340 pixels (across-orbit × along-orbit). The spatial resolutions corresponding to the pixels were similarly merged, with smaller spatial resolutions in regions closer to the imager and larger spatial resolutions in regions farther from the imager. Using the principles of similar triangles, the minimum spatial resolution was calculated to be 1.2 × 2.4 km, while the maximum spatial resolution was 6.7 × 5.3 km.

Detector noise includes the photon shot noise, dark current noise, and readout noise. The photon shot noise is related to the number of photons reaching the detector pixels, denoted by $\sigma_s = \sqrt{N_{sca}}$. The dark current noise is independent of the signal level and is associated with the performance of the detector itself, with a value of 10 e-/(s·pixel). The readout noise arises from electronic processes during the transfer of charge from the pixel out of the camera, with a value of 3.9 e-/(s·pixel). All three noise types were treated as Gaussian noise, indicating that their probability density functions followed a Gaussian distribution.

**3 Three-dimensional reconstruction of PMCs**

**3.1 Data**

The AIM satellite works in a sun-synchronous orbit with an approximately noon local time equator crossing. The CIPS instrument employs four UV imaging cameras with a wide field of view centered in the nadir. CIPS is a four-camera UV imager that makes hemispheric-scale measurements of PMCs. The details of the instrument performance and the data quality

were given by Carstens et al. (2013) and Lumpe et al. (2013). The strength of CIPS is to provide the PMCs maps (occurrence





frequency, albedo, IWC, and particle radius) with high horizontal resolution and coverage of summer polar. This work uses the CIPS level 2 orbit IWC maps, in which one day includes 15 rectangular images for 15 orbits respectively. Data in this level provides a higher-level summary and gives an averaged spatial resolution of 5×5 km$^2$ throughout the orbit strip. In this work, we used seven PMCs seasons in north hemisphere during the years 2008~2014.

SOFIE performs satellite solar occultation measurements to determine vertical profiles of PMCs' properties as well as the surrounding temperature, pressure, and abundance of H$_2$O. It observes 15 sunrise solar occultations at a latitude range of 65° N to 86° N in the NH and 15 sunset solar occultations at latitudes from 63° S to 78° S in the SH. The field of view (FOV) is about 1.5 km vertical × 4.3 km horizontal. Detectors are sampled at 20 Hz, which corresponds to ~145m vertical spacing, or roughly 10 times over-sampling. The line-of-sight (LOS) through the PMCs layer at 83 km is ~290 km. The data from Version

1.3 with more precise data are used in this work, in which the profiles have the finest vertical resolution of less than 0.2 km (Hervig et al., 2009; Marshall et al., 2011; Stevens et al., 2012).

### 3.2 Reconstruction method

The 3D structural information of PMCs serves as a crucial input background field for the forward model. To reconstruct a 7-year PMCs 3D field, a combination of CIPS horizontal 2D cloud field data and SOFIE vertical profile data was used. The IWC

played a crucial role in establishing the correspondence between cloud microphysical parameters.

Statistical analyses were conducted on the data of IWC, ice mass content ($M_{ice}$), effective radius ($r_e$), and particle size spectrum width ($d_r$) in the Northern Hemisphere detected by SOFIE carried by the AIM satellite from 2008 to 2014. Interestingly, the distributions of $M_{ice}$, $r_e$, and $d_r$ corresponding to different IWC values conform to a Gaussian distribution, and the fitting results are as follows:

$$M_{ice}(h') = a_m e^{\frac{-(h'-b_m)^2}{2c_m^2}} \qquad (13)$$

$$r_e(h') = a_{re} e^{\frac{-(h'-b_{re})^2}{2c_{re}^2}} \qquad (14)$$

$$d_r(h') = a_{dr} e^{\frac{-(h'-b_{dr})^2}{2c_{dr}^2}} \qquad (15)$$

where $h'$ is the altitude. The distribution of $a_i$ values follows the power function distribution, while the distribution of $b_i$ and $c_i$ values conforms to the exponential function distribution. The data were fitted, and the resulting image is shown in Fig.

5. The fitted functions are listed in Table 1.

Table 1 Fitted expressions for the ice mass content, effective radius, and particle size spectrum width were obtained.

| | $a_i$ | $b_i$ | $c_i$ |
|---|---|---|---|
| $M_{ice}$ | $0.094 \times IWC^{1.191}$ | $1.616 \times e^{-\frac{IWC}{70.700}} + 82.856$ | $1.010 \times e^{-\frac{IWC}{61.037}} + 1.417$ |





| | | | |
|---|---|---|---|
| $r_e$ | $2.215 \times \text{IWC}^{0.521}$ | $1.166 \times e^{-\frac{IWC}{30.226}} + 82.875$ | $0.817 \times e^{-\frac{IWC}{26.044}} + 1.905$ |
| $d_r$ | $0.933 \times \text{IWC}^{0.482}$ | $1.069 \times e^{-\frac{IWC}{25.169}} + 83.049$ | $0.867 \times e^{-\frac{IWC}{25.006}} + 1.859$ |

In most previous studies on PMCs, the particle size distribution has been described as a log-normal distribution (Thomas et al.,1985). The log-normal distribution is described by the concentration, effective radius, and distribution width. Rapp et al. (2006) recently demonstrated that PMCs particle size distribution can be more accurately represented using a Gaussian

distribution, which is described by the concentration, effective radius, and particle size spectral width. Each altitude at each IWC corresponds to a scale spectral parameter, $N_i$:

$$N_i(r) = a_N e^{\frac{-4ln2(r-r_e)^2}{d_r^2}} \qquad (16)$$

The distributions of $r_e$ and $d_r$ have notable regularity, which can be obtained by fitting, whereas the distribution of the concentration ($a_N$) shows less regularity.

Assuming that the axis ratio of the ice particles is 2 and the volume of a single ice particle is $4/3\pi ab^2$, where $a$ is the long axis and $b$ is the short axis ($a:b = A_R = 2$), the radius ($r$) is given by $(a+b)/2$. Subsequently, the volume of a single PMCs particle was obtained as $V = 64/81\pi r^3$. As the ice particle has a density of 0.92 g/cm³, the mass of a single particle can be expressed as $m = \rho V$. Accordingly, the mass of $N_i$ particles is given by $m_i = \rho V \times N_i$. The concentration can be calculated as follows:


$$a_N = \frac{M_{ice}}{\sum_{r=5}^{r=100} \frac{10^{-21} \times 0.92 \times 64\pi r^3}{81} e^{\frac{-4ln2(r-r_e)^2}{d_r^2}}} \qquad (17)$$

Based on the IWC measured at each location by the CIPS, the profiles of the ice mass content and scale spectral distribution at each altitude can be derived, which leads to the structure of the 3D cloud field. When the IWC is 100 g km⁻², the scale spectrum distribution can be plotted for the height range of 75–85 km at 0.2 km intervals (see Fig. 4g–i).



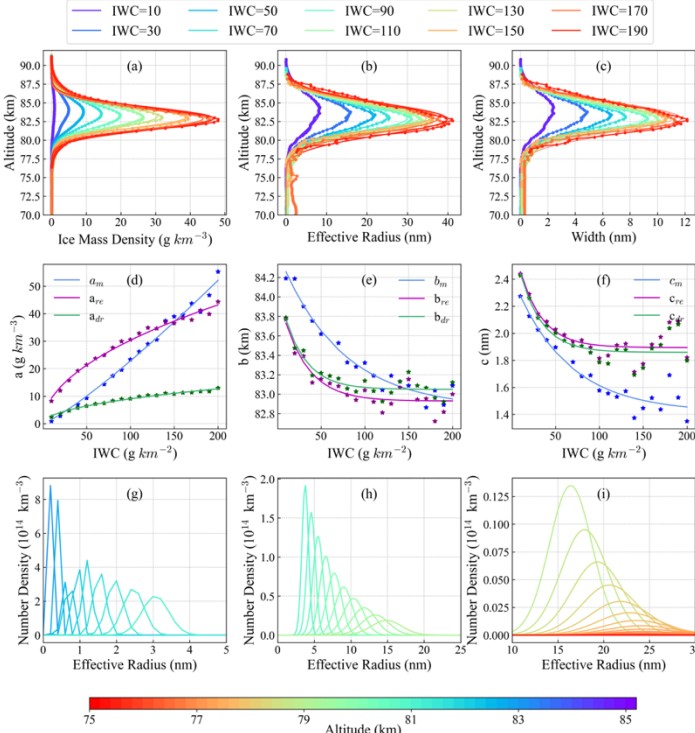

**Figure 4: (a)-(f) Data fitting of the ice mass content, effective radius, and particle size spectrum width. (g)-(i) Scale spectrum distribution at different heights (75–85 km) when IWC = 100 g km$^{-2}$.**

The results in Fig. 5 present an illustrative example from the CIPS data on August 8, 2009, for the 10[th] orbit. Based on the IWC at each location (Fig. 5a), we obtained the IWC, effective radius, and particle size spectral width at different altitudes at each location. For the altitude range of 81–85 km, we derived the distributions of the ice mass content, effective radius, and particle size spectral width at 1 km intervals, as shown in Fig. 5b–d, respectively. The results in Fig. 5e–g represent the profiles taken at a latitude of 70° N. For this orbit, the PMCs near 70° N were more prominent and exhibited higher intensity. Both the 3D images and profiles indicate that the peaks of the PMCs were mainly present at an altitude of approximately 83 km.

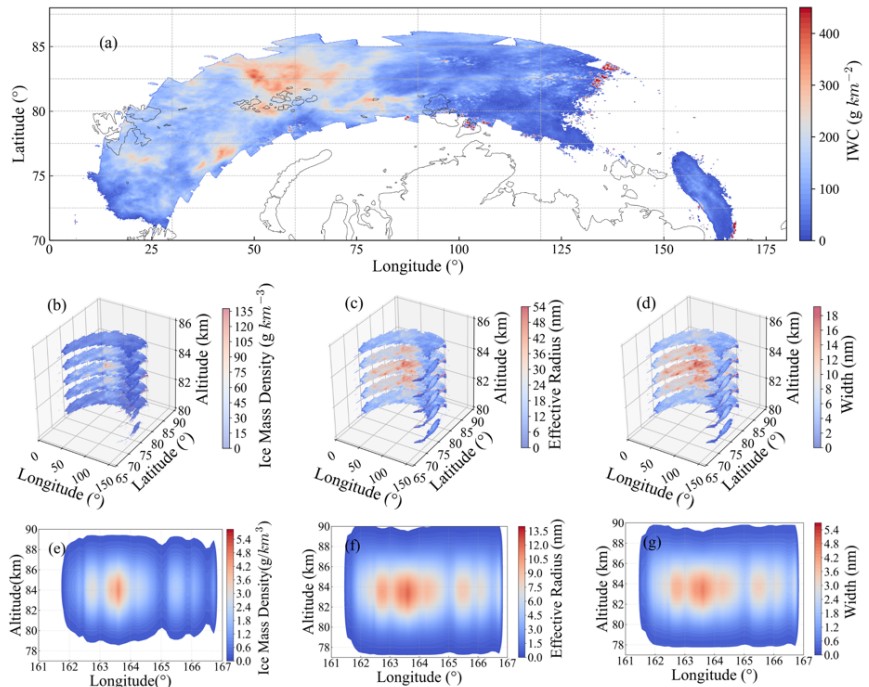

**Figure 5: (a) IWC distribution of the 10th orbit on August 8, 2009. (b)–(d) 3D diagrams of $M_{ice}$, $r_e$ and $d_r$ at different heights. (e)–(g) Changes in $M_{ice}$, $r_e$ and $d_r$ with altitude and latitude.**

## 4 Simulation result

### 4.1 Instrument Description and Parameters

Unlike CIPS, which employs four UV CCD cameras, WFUI employs only one UV CCD camera, resulting in a smaller payload size, reduced power supply, and diminished data transmission requirements. This design allows greater flexibility in installing satellites that are not specifically dedicated to PMCs detection, such as remote sensing and meteorological satellites. The CCD camera within the WFUI captures scattered light from various regions of the PMCs and converts it into electronic signals for recording. The spatial area covered by each pixel of the CCD camera on the image sensor was determined by its FOV in the optical system and the satellite altitude. The focal length of the ultraviolet imaging camera determines the FOV of WFUI. FOV = $2\xi \times \xi_0/1,000/f$ /2, where $\xi$ denotes screen resolution, $\xi_0$ denotes single pixel size. The filter constitutes another essential optical element featuring a chosen central wavelength of 265 nm and a full width at half maximum (FWHM) of 15 nm. By applying the filter parameters to Eq. (6), the number of photons received per second per unit area of altitude of the PMCs was derived. The WFUI is designed to be integrated into sun-synchronous satellites that fly approximately 15 orbits per day, each of which can capture elongated images. In practice, each elongated image of an orbit comprised 25 individual images captured using a CCD camera. Each orbit took approximately 20 min to achieve coverage at latitudes greater than 50 °, with an average horizontal spatial resolution of 5 × 5 km.



Table 2 Input parameters for the forward model of the WFUI

| Unit | Parameter | Symbol | Reference Value |
|---|---|---|---|
| **Satellite** | Mean altitude | $H$ | 600 km |
| | Mean orbital velocity | $V$ | 7.56 km/s |
| | Spacecraft view angle | $\omega$ | 30 ° |
| **Filter** | Central wavelength | $\lambda_0$ | 265 nm |
| | Filter FWHM | FWHM | 15 nm |
| | Refractive index of glass base | $n_e$ | 2.0 |
| | Peak transmittance | $t_{fmax}$ | 0.6 |
| **CCD Camera** | Measurement cadence | $T_M$ | 43 s |
| | Exposure times | $T_e$ | 0.75 s |
| | Focal length | $f$ | 23.5 mm |
| | CCD Diagonal | $D_C$ | 38 mm |
| | Iris | $F$ | 1.625 |
| | FOV | $\Delta\theta$ | 60 ° |
| | Single pixel size | $\xi_0$ | 20 um × 20 um |
| | Screen resolution | $\xi$ | 1360 × 1360 |
| | Quantum efficiency | $\eta$ | 0.80 |
| | Analog-to-digital unit | $C_{ADU}$ | 7 |
| | Number of pixel column | $\mu$ | 16 bit |

**4.2 Simulation results for orbits**

The geographical data required for simulating the detector signal of a single orbit were sourced from CIPS level 2 data, encompassing latitude, longitude, local time, solar zenith angle, and other relevant information. Based on these data, the number of photons, $N_{ph}$, per second per unit area of the WFUI at each position throughout the orbit can be calculated.

Taking the first orbit on August 3, 2011, as an example, the process for detecting PMCs by WFUI was simulated. Figure 6c shows the IWC detected by the CIPS. Assuming that the PMCs detected by CIPS reflected the actual conditions, a simulated IWC corresponding to a single CCD camera shot was obtained (Fig. 6a). The number of photons at the respective positions was determined by calculating the optical characteristics and radiative transfer. The number of photons captured by the WFUI in a single shot was obtained by calculating the coverage of a single image according to each imager parameter (Fig. 6b) at a spatial resolution of 5 × 5 km. The number of photons was then determined based on the calculated resolution of each CCD pixel, yielding the distribution shown in Fig. 6d, which represents the number of photons for 170 × 340 individual pixels. Using the formula from Section 2.7, the gray values for 170 × 340 pixels were computed; after error correction, the gray values were obtained for Fig. 6e. The detectable information of the PMCs was derived by subtracting the noise twice from the data in Fig. 6e (Fig. 6g). The region of 170 × 340 pixels was subdivided into 1,700 × 3,400 pixels, with the gray value of each pixel reduced to 1/100 of its original value. Subsequently, the spatial resolution was downsized by a factor of 10 in both $P_W$ and $P_H$. Subsequently, these adjusted values were merged to achieve a 5 × 5 km spatial resolution, aligning with the latitude and longitude data from CIPS, thus simulating the detected PMCs on the map (Fig. 6h). Figure 6f illustrates the simulated PMCs captured along the orbit of the AIM satellite using the simulated WFUI. Figure 6i was obtained after denoising.



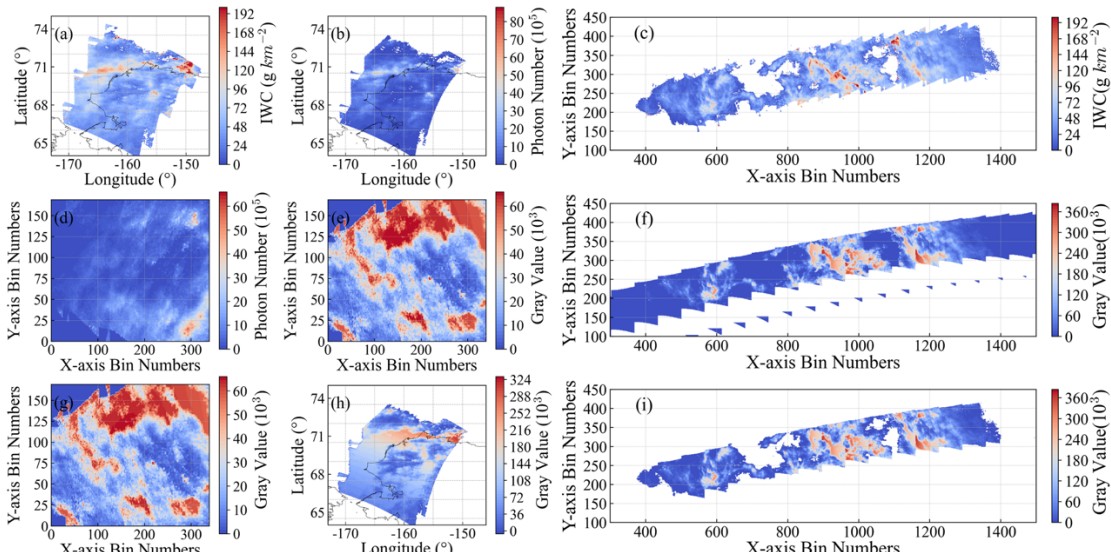


**Figure 6: (a) Distribution of IWC with latitude and longitude for a single image detected by CIPS. (b) Distribution of the Photon number with latitude and longitude for a single image detected by WFUI. (c) IWC of the orbit detected by CIPS. (d) Photon number corresponding to the single image pixels. (e) Gray value corresponding to the single image pixels. (f) The gray value of the orbit detected by WFUI. (g) Gray value of the single image after the denoising process. (h) Distribution of gray value with latitude and longitude for the single image after the denoising process. (i) Gray value of the orbit after the denoising process of WFUI.**

Comparing Fig. 6c with 6i, the WFUI cannot detect all PMCs, and its detection efficiency is less than 100 %. Owing to the stochastic nature of noise, information regarding the PMCs may be obscured. If the gray value remains greater than zero after subtracting the noise remains greater than 0, there are PMCs at that location. The detection efficiency is calculated as the ratio of the number of data points with detected PMCs in the simulated signal to the number of data points with PMCs in the input

PMCs field. Taking the first orbit on August 3, 2011, as an example in Fig.6 for the calculation, the detection efficiency of PMCs by the WFUI was determined to be 81.64 %:

$$\text{Detection Efficiency} = \frac{\text{Number of data with simulated signals greater than zero}}{\text{Number of data with positive IWC in the input PMCs field}} \quad (18)$$

We note that the construction of the PMCs 3D field relies primarily on a diverse range of data provided by SOFIE. The SOFIE exhibits excellent sensitivity and detection capability for smaller ice particles. However, owing to the constraints posed by the

detection principles, instrument characteristics, and orbital factors of the CIPS, its sensitivity to small particles is comparatively limited. Consequently, when employing the IWC of the PMCs detected by CIPS as the input background, the spatial structure of the reconstructed PMCs 3D field was essentially derived from the PMCs structures detected by SOFIE. The resulting simulated signals were based on the detection data from either the CIPS or WFUI. One outcome of this approach is the potential for the WFUI to be unable to detect PMCs when the input PMCs field exhibits a lower IWCs.

On August 3, 2011, CIPS had 15 orbits covering the Northern Hemisphere. The IWC from these 15 orbits were combined to form the background distribution of PMCs for that day, as shown in Fig. 7a. Following the processing steps outlined in Fig. 6





for simulating the detection of PMCs by the WFUI, data from 15 orbits were processed to obtain gray values by removing noise. Combining the gray values from these 15 orbits results in information on the PMCs detected by the WFUI, as shown in Fig. 7b.

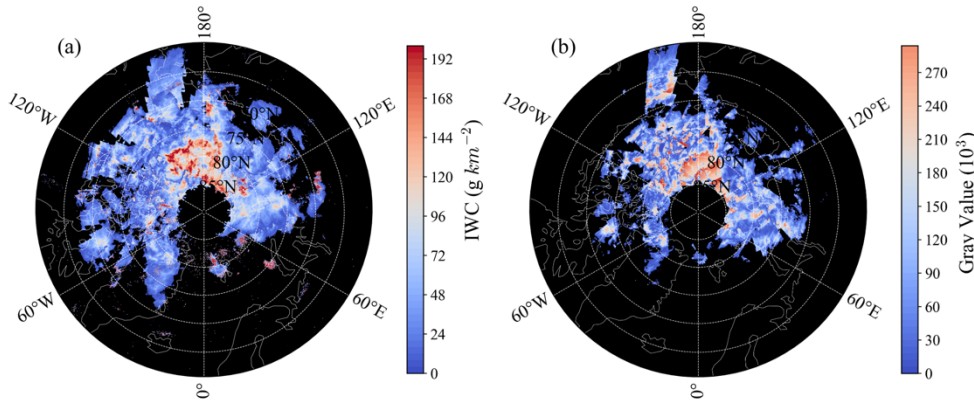


**Figure 7: (a) Distribution of IWC on August 3, 2011, detected by CIPS. (b) Simulating distribution of gray value.**

**4.3 Detection efficiency analysis for PMCs seasons**

The geographical data required for simulating the detector signal for the PMCs season were sourced from the CIPS level 3 data, encompassing latitude, IWC, solar zenith angle, and other relevant information. Based on these data, the number of
photons, $N_{ph}$, per second per unit area of the WFUI for the PMCs season can be calculated. The processing method for Level 3 data was essentially the same as that for Level 2 data. As the level 3 data are the overall results after averaging, there is no need for image calibration, such as phase shift and flat field. The resolution was 5 × 5 km, and the detection efficiency per orbit during the PMCs season was calculated.

Statistical analysis of the IWC and detection efficiency from 2008 to 2012 revealed daily average variations. Given that PMCs
often appear in the Northern Hemisphere from mid-May to August, around the time of the summer solstice, the timescale is described in terms of days relative to the summer solstice. As shown in Fig. 8a–b, PMCs are scarce, with a relatively low IWC at the beginning of the PMCs season, making them susceptible to being overwhelmed by noise, resulting in lower WFUI detection efficiency. As the PMCs season progressed, the occurrence of PMCs gradually increased, accompanied by a higher IWC. Noise becomes insufficient to mask the PMCs information, leading to an increase in the detection efficiency.

Furthermore, statistical analysis of data from 2008 to 2012 allowed the examination of variations in the IWC and detection efficiency with latitude in the Northern Hemisphere. As depicted in Fig. 8c–f, the IWC of PMCs at lower latitudes is relatively low. Owing to limited data availability in lower-latitude regions, the amplitude of the IWC fluctuations with latitude is relatively large, with no prominent trend. However, in general, the IWC tends to increase with increasing latitude, along with a corresponding increase in detection efficiency. A similar trend was observed in the Southern Hemisphere, with both the IWC

and detection efficiency increasing with increasing latitude. Additionally, in the southern or northern hemisphere, the IWC
tends to remain stable beyond 80° latitude, and the detection efficiency remains generally constant or even slightly decreases.

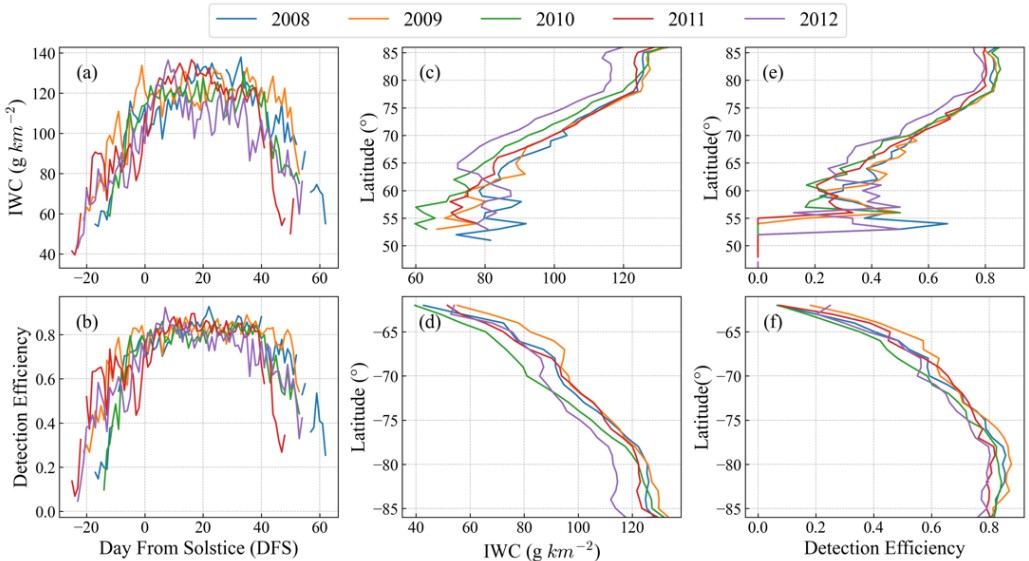

**Figure 8: (a) Average daily variation of IWC in PMCs season from 2008 to 2012. (b) Average daily variation of detection efficiency in PMCs season. (c) Variation of IWC with latitude in the northern hemisphere during the PMCs season from 2008 to 2012. (d)**
**Variation of IWC with latitude in the southern hemisphere. (e) Variation of detection efficiency with latitude in the northern hemisphere. (f) Variation of detection efficiency with latitude in the southern hemisphere.**

By fitting the correlation curve between the detection efficiency, $\eta_d$, and IWC from 2008 to 2012, the resulting relationship
is given by:

$$\eta_d = -2.325 \times e^{-\frac{IWC}{54.162}} + 1.068 \qquad (19)$$

From Fig. 9, there is a strong correlation between the IWC of the PMCs and the detection efficiency, with the numerical
distribution following an exponential function pattern. When the IWC was relatively low, the detection efficiency increased
rapidly as the IWC increased. However, once the IWC reached a higher level, the rate of increase in the detection efficiency
decreased with further increments in the IWC.





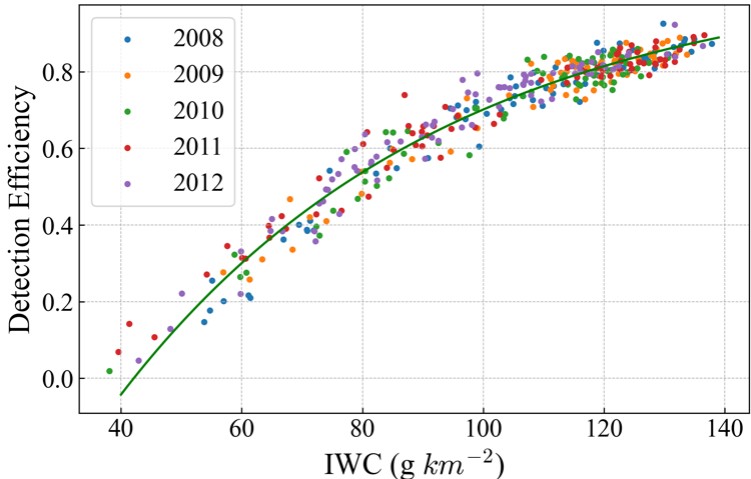

**Figure 9: Correlation between the IWC and detection efficiency in the PMCs season for both hemispheres.**

## 4.4 Sensitivity analysis of main parameters

A parameter sensitivity analysis is necessary to examine the influence of various simulated parameters on the WFUI simulation results and select the appropriate parameters more precisely to achieve enhanced PMCs detection and improved detection efficiency. This analysis alters the FOV of the WFUI and the satellite altitude to observe variations in the detection efficiency. The FOV of the WFUI could be adjusted by modifying the focal length. As the FOV increased, the pixel resolution of areas closer to the WFUI changed minimally, whereas the pixel resolution of farther areas increased. The area captured by the WFUI in a single shot forms an isosceles trapezoidal shape. As the FOV expands, the two base angles of the trapezoid decrease, thereby increasing the coverage area. Conversely, with a smaller FOV, the range detected by the WFUI becomes limited, preventing complete coverage of the simulated PMCs field and reducing detection efficiency. Enlarging the FOV aids in capturing the simulated cloud field more effectively, leading to improved detection efficiency. Furthermore, increasing the CCD pixel resolution amplified the strength of the PMCs signal, making it less sensitive to interference from noise signals, thereby enhancing the detection efficiency. Increasing the FOV further improves the detection efficiency. However, an excessively large FOV may lead to the capture of areas where a significant presence of PMCs is unlikely, resulting in resource wastage. Additionally, when the CCD pixel resolution is excessively high, the accuracy of PMCs detection may decrease. While altering the satellite altitude, factors such as satellite velocity, photon reception efficiency, and CCD pixel resolution change. When maintaining other parameters constant, a higher orbital altitude results in a lower $R_{at}$ and a higher CCD pixel resolution. Given the more substantial variation in the CCD resolution, its impact on the gray values becomes more pronounced, inevitably leading to an increase in the PMCs detection efficiency. However, an excessively high CCD pixel resolution may not be favorable for detecting PMCs. Therefore, when adjusting the satellite altitude, we must modify the FOV of the WFUI concurrently to ensure that the change in the covered area between parameter variations remains moderate.





In this study, AIM satellite data from the 2009 PMCs season were used, spanning 114 days and encompassing a total of 1,756 orbits. To reduce computational time, one orbit was randomly selected each day for sampling. Specifically, scenarios were considered where the satellite altitude was 600 km, with the FOV of the WFUI ranging from 54 ° to 66 °, as well as cases with an FOV of 60 ° and satellite altitudes ranging from 500 to 800 km. Furthermore, the detection efficiency of the PMCs was

analyzed for situations in which the satellite orbital altitude and imaging instrument FOV were 500 km/67 °, 550 km/63 °, 600 km/60 °, 650 km/57 °, 700 km/53 °, 750 km/50 °, and 800 km/48 °. For a comprehensive analysis, the entire PMCs season was divided into five periods, with 15 days before and after as the early and late stages, respectively; the remaining data-forming periods were 27 days each, namely the early, mid, and mid-late periods. The results presented in Fig. 10 were obtained by analyzing the data from these five periods. During the early and late periods of the PMCs season, when PMCs occurrence was relatively low, variations in the parameters had a smaller impact on the detection efficiency. However, in the early, mid,

and mid-late periods, when PMCs occurrence was higher, the effect of increasing satellite altitude on detection efficiency was clearly evident. Simultaneously, an observable trend of increased detection efficiency was also apparent with the enlargement of the FOV of the WFUI.

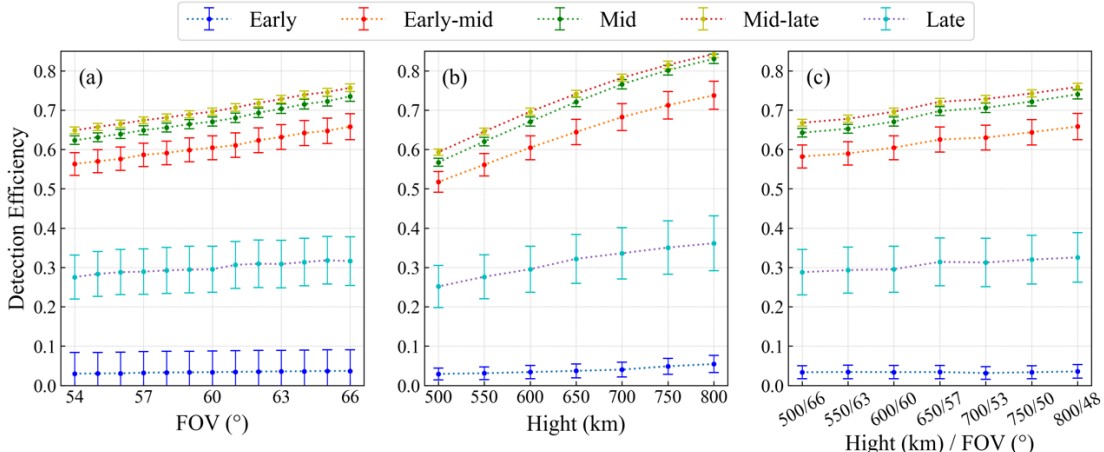

**Figure 10: (a) Variation in detection efficiency with the FOV at different times; (b) variation in the detection efficiency with satellite altitude; and (c) variation in the detection efficiency with simultaneous changes in the FOV and satellite altitude.**

## 5 Conclusion

The variation trend and characteristics of PMCs have significance for studying the evolution of atmospheric systems and understanding various atmospheric dynamic processes. UV imaging technology has been proven to have good application

prospects in detecting PMCs. In this study, we proposed the WFUI and built a forward model to evaluate the detection capability and efficiency. Fusion of CIPS and SOFIE data was employed to reconstruct the 3D PMCs scene as the input background. Based on calculations of the optical properties, radiative transfer, and detection processing, the actual detection



signal was simulated by photoelectric conversion. A comparison with the input background field was conducted to compute and analyze the detection efficiency. The results have shown that:

(1) In the initial duriations of the PMCs season, the relatively low IWC within PMCs leads to a constrained detection efficiency for the WFUI. As the PMCs season progresses, the IWC of PMCs gradually increases, resulting in a significant increase of the detection efficiency. The relationship between IWC and detection efficiency follows an exponential function distribution.

(2) As the latitude increases, there is an observable upward trend in the IWC of PMCs, which directly influences the variations
in detection efficiency. Once latitude surpasses 80 degrees, IWC stabilizes, and the detection efficiency at this time also remains basically unchanged, or even slightly decreased.

(3) The increase of satellite altitude will improve the detection efficiency. In addition, an increase in the FOV of the WFUI will also lead to an increase in detection efficiency.

The WFUI instrument primarily consists of a CCD camera and a lens,which is designed to be compact and allows for greater
flexibility in installation on satellites not specifically dedicated to detecting PMCs. According to the estimation of the instrument parameters shown in Table 2, the size of the entire instrument is around 3 L, and it is relatively lightweight. This allows for the possibility of deploying the instrument on a 3-unit CubeSat.

**Data availability**

The SOFIE PMC data are now available to the public in the form of summary files containing data for each PMC season at
http://sofifie.gats-inc.com/sofifie/index.php. The CIPS/AIM PMC data are now available to the public in the form of summary files containing data for each PMC season at http://lasp.colorado.edu/aim/download-data.php.The NRLMSISE-00 Atmosphere Model is available at https://ccmc.gsfc.nasa.gov/modelweb/models/nrlmsise00.php.

**Author Contributions**

Conceptualization: Haiyang Gao
Formal analysis: Ke Ren,  Haiyang Gao
Methodology: Ke Ren, Haiyang Gao, Shuqi Niu
Resources: Leilei Kou, Lingbing Bu, Shangyang Sun
Validation: Ke Ren
Visualization: Ke Ren
Writing -original draft: Ke Ren
Writing-review & editing: Haiyang Gao



**Competing interests**

The contact author has declared that none of the authors has any competing interests.

**Acknowledgments**

The authors acknowledge funding from the National Key Research and Development Program of China (No. 2021YFC2802502). The financial support by the National Natural Science Foundation of China (42374223) and the Industry University Research Cooperation Fund of the Eighth Research Institute of China Aerospace Science and Technology Corporation are also acknowledged.

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
