# Peer review of "Simulation and detection efficiency analysis for polar mesospheric clouds measurements using a spaceborne wide field of view ultraviolet imager"

_Atmospheric Measurement Techniques, 2023_

## Author Comment (AC1)

Ke Ren and Haiyang Gao
School of Atmospheric Physics
Nanjing University of Information
Science and Technology
Nanjing, 210044, P. R. China
E-mail: renke0412@gmail.com

June 10th, 2024

Re:      Manuscript Number: AMT-2023-186
Title:    Simulation and detection efficiency analysis for polar mesospheric clouds measurements
            using a spaceborne wide field of view ultraviolet imager
Author: Ke Ren; Haiyang Gao; Shuqi Niu; Shaoyang Sun; Leilei Kou; Yanqing Xie; Liguo Zhang;
            and Lingbing Bu

Dear Reviewer,

Thank you for your review of the manuscript entitled "Simulation and detection efficiency analysis for polar mesospheric clouds measurements using a spaceborne wide field of view ultraviolet imager" as well as for your valuable suggestions to improve the paper. There may be some procedural mistakes in the reviewing process. I'm pleased to hear that you found the revisions satisfactory.
I'm posting here again the modifications made based on your suggestions. Please note that the reviewer's comments are shown in bold type and our responses in plain type.

Best regard,
Ke Ren and Haiyang Gao

**Summary:**

**Polar mesospheric clouds (PMCs) are critical for modulating polar and global atmospheric processes. They were previously observed mainly by the CIPS instrument onboard the satellite AIM, which based on the UV imaging technology. However, there is no future plan launching similar instruments into space, causing a gap in monitoring PMCs.**

**The authors of this paper proposed a new spaceborne wide FOV UV imager (WFUI), and conducted simulation studies to assess the performance of this newly designed imager. Results show that the WFUI performed well in PMCs detection, verified by high detection efficiency. They also found that the detection efficiency varies with seasons due to the seasonal variation of ice water contents in PMCs. Additional sensitivity studies show how to further optimize the WFUI to improve its detection efficiency.**

**This work by the authors is interesting. Their presentation is thorough and comprehensive. Findings from this work are important. However, there are some places where more clarifications are needed, therefore I suggest a moderate revision for this version of draft.**

**Major comments:**

**Questions about methodology:**

**1. L94: "This absorption is primarily from 20 to 25km": it would be more informative to show how the absorption (or transmittance) changes with altitude based on your simulation. Please add the following figure or similar:**

**X-axis: wavelength, Y-axis: altitude, color: absorption or transmittance.**

**In this way, you can show readers the whole vertical structure of absorption covering the 265nm band.**

> Thanks for the reviewer's good comment and instruction.
>
> We utilized the LBLRTM model to compute the transmission at different altitudes for a wavelength of 265 nm, providing readers with a more intuitive representation of the entire vertical absorption structure at 265 nanometers, as illustrated in Figure 1d.

[Figure]

**2. Equation for Solar radiation: several symbols are not explained:**

**(1) \delta_\lambda and its value, (2) t_{\lambda}, (3) c_1, c_2, (4), t_{fmax}, 5, FWHM**

> Thanks the reviewer for pointing this out.
>
> The symbols mentioned above are explained in Table 2, 'Input parameters for the forward model of the WFUI', in Section 4.1. However, the separation between Table 2 and the current

formula may create difficulties in understanding. This was an oversight on our part. To address this, we have supplemented explanations for all symbols after the Equation for Solar radiation.

**3. RT calculation: L181-182: "The altitude range….": It would be better if you can add a figure showing the vertical distribution of your atmospheric levels, with corresponding heights and approximated pressures.**

Thanks for the reviewer's good comment and instruction.

We have added Figure 1e, illustrating the vertical distribution of atmospheric levels with corresponding heights and approximated pressures. In the text, we provide an explanation for adopting this atmospheric segmentation. The text is revised as follows:

"... *Due to the presence of PMCs within the altitude range of 75 km to 85 km, the altitude range of 75–85 km is meticulously segmented into 50 layers, each with intervals of 0.2 km. The altitude range of 40–75 km was divided into seven layers at 5 km intervals; and 85–120 km was divided into seven layers at 5 km intervals, resulting in a total of 64 layers (Fig. 1e). This approach to altitude layer segmentation not only maintains the accuracy of the calculation results but also effectively improves computational efficiency. ...*"

[Figure]

**Questions related to results:**

**L324: "This design allows greater flexibility…meteorological satellites": since now building cubesat constellation is gaining popularity, is it possible to install WFUI on those cubesats? How large/weight will your instrument be? If it is possible, it would be great if the authors can add additional discussion for this part in the Summary section, since I believe this will increase the impact of this paper.**

Thanks for the reviewer's suggestion. We add additional discussion for this part in the Summary section.

"... *The WFUI instrument primarily consists of a CCD camera and a lens,which is designed to be compact and allows for greater flexibility in installation on satellites not specifically dedicated to detecting PMCs. According to the estimation of the instrument parameters shown in Table 2, the size of the entire instrument is around 3 L, and it is relatively lightweight. This allows for the possibility of deploying the instrument on a 3-unit CubeSat. ...*"

**Minor comments:**

**L14: "such as the Cloud Imaging": "such as" => "adopted by": because the first half of this sentence is the technology, while the second half is about research.**

**L15. "and development of PMCs": remove "and developments of": PMCs are the studying subjects.**

Thanks for pointing this out. We have rectified the semantic error in this sentence.

"*… Ultraviolet (UV) imaging technology, adopted by the Cloud Imaging and Particle Size (CIPS) instrument onboard the Aeronomy of Ice in the Mesosphere (AIM) satellite, has significantly advanced the research on PMCs. …*"

**L29: "revealed an exponential relationship … of PMCs": between the IWC and what?**

Thanks for pointing this out. We have revised it to 'an exponential relationship between the ice water content (IWC) of PMCs and detection efficiency'.

**L40: "and indicator of long-term changes in the Earth's atmosphere": long-term changes of what? Please be more specific**

Thanks for pointing this out.

The term 'Indicator of long-term changes in the Earth's atmosphere' specifically denotes long-term changes in temperature and water vapor content. We appreciate your clarification, and the revised information has been incorporated to enhance precision.

"*… Thus, the long-term trend in PMCs variation is considered an indicator of long-term changes of temperature and water vapor content in the Earth's atmosphere. …*"

**L41: "Recently, the frequency": It would be better to write the approximate time (e.g., year, month, day) here since "Recently" have different meanings for atmospheric researchers studying different phenomenon.**

Thanks for the reviewer's suggestion.

We have updated the statement to provide a more specific timeframe. Additionally, for further details and comprehensive information, please refer to [Miao et al., 2022; Kaifler et al., 2018; Tylor et al., 2017].

"*… Prior research has indicated that the frequency and obse*
*rved brightness of PMCs in the mid-latitudinal areas have increased over the past half-century. …*"

**L53: "Himwari-8": which sensor on Himawari-8 do you refer to?**

Thanks for the reviewer's good comment and instruction.

'Himawari-8' referred to denotes the Advanced Himawari Imager (AHI) sensor mounted on the Himawari-8 satellite. This information is derived from the study conducted by Tsuda et al. in 2018. Tsuda et al. (2018) provide an initial report on PMCs observations by the satellite Himawari-8. AHI can produce full-disk images including the Earth's limb, which would provide valuable opportunities for PMC observations by continuous limb-viewing from its almost fixed location relative to the Earth.

**L445: Figure 10: Should it be "Hight" instead of High**

Thanks for pointing this out.

We have replaced 'high' with 'height' in this Figure 10.

[Figure]
* * *
Special thanks to the reviewer for his/her good comments.
* * *
We have tried our best to revise and improve the manuscript and made a few changes in the manuscript according to the reviewer's good comments. Again, many thanks for your valuable comments and suggestions. We would like to have our paper at your disposal.

We appreciate for reviewer's spending more time on reviewing over paper and offering valuable suggestions and hope that the modification and corrections will meet with approval and hoping that our paper will be published in *Atmospheric Measurement Techniques* as soon as possible.

We look forward to your information about my revised paper.

Yours sincerely,
Ke Ren and Haiyang Gao

---

## Author Comment (AC2)

Ke Ren and Haiyang Gao
School of Atmospheric Physics
Nanjing University of Information
Science and Technology
Nanjing, 210044, P. R. China
E-mail: renke0412@gmail.com

June 10th, 2024

Re:      Manuscript Number: AMT-2023-186

Title:    Simulation and detection efficiency analysis for polar mesospheric clouds measurements
          using a spaceborne wide field of view ultraviolet imager

Author: Ke Ren; Haiyang Gao; Shuqi Niu; Shaoyang Sun; Leilei Kou; Yanqing Xie; Liguo Zhang;
          and Lingbing Bu

Dear Reviewer,

Thank you for your review of the manuscript entitled "Simulation and detection efficiency analysis for polar mesospheric clouds measurements using a spaceborne wide field of view ultraviolet imager" pending revision as well as for your valuable suggestions to improve the paper. We have tried every means possible to improve the presentation and increased the readability. We have also made all the changes suggested by the reviewer and addressed all the comments in the notes below. Please note that the reviewer's comments are shown in bold type and our responses in plain type.

Best regard,
Ke Ren and Haiyang Gao

**GENERAL COMMENTS:**

**This paper presents an analysis of anticipated performance for a wide field-of-view ultraviolet imager (WFUI) instrument designed to observe polar mesospheric clouds (PMCs). The WFUI is based on the Cloud Imaging and Particle Size (CIPS) instrument flown on the Aeronomy of Ice in the Mesosphere (AIM) satellite, but with only one forward-looking camera instead of four cameras. A database of expected PMC behavior and sFor Ice Experiment (SOFIE) instruments. This database is used in a forward model to simulate the performance of the WFUI for a wide range of conditions. The likelihood of successfully detecting a PMC is shown to be dependent on ice water content, which has implications for the results available at different latitudes and different times during a PMC season.**

**The paper is well-written and comprehensive. Selected comments are listed below.**

**SPECIFIC COMMENTS**

**1. Page 3, lines 75-76: I would clarify this statement to say that CIPS is the only nadir view instrument to use multiple views of the same location and phase function effects to identify PMCs. SBUV-type instruments also use nadir viewing and UV wavelengths, but with only one large pixel.**

> Thanks for the reviewer's good comment. We appreciate your clarification, and we have modified this sentence (lines 75-76) as:
>
> "...*CIPS is the only nadir view instrument to use multiple views of the same location and phase function effects to identify PMCs. SBUV-type instruments also use nadir viewing and UV wavelengths, but with only one large pixel...*"

**2. Page 4, lines 100-101: This operating concept sounds similar to CIPS, except that CIPS uses 4 cameras (forward, backward, left side, right side) to get multiple views of each location at different scattering angles (SCA). The large variation in phase function with SCA for small ice particles then enables identification of pixels containing PMCs. Since the WFUI instrument will only capture 1 (or perhaps 2) images of a given location with much less SCA variation, how will PMCs be identified relative to the background signal?**

> Thanks for pointing this out. We have discussed this issue, and the information has been incorporated.
>
> As mentioned by the reviewer, CIPS has four cameras, which gives it the ability to detect weak PMCs by examining the phase function from SCA with significant differences. However, by an intermittent exposure, the WFUI instrument can capture two (or more) images of a given location, but with a scattering angles (SCA) variation of only 20-40 degrees. This indeed makes it challenging to identify weaker PMCs, which is one of the reasons the lower detection efficiency in regions of weaker PMCs. However, using a single camera significantly reduces costs and minimizes payload space. Also, WFUI is potential to be installed on multiple CubeSats to detect the same sampling PMCs region with variation scattering angle when different CubeSats are orbiting in the same orbit with a certain delay time interval. This would allow obtaining multiple views of each position from different SCA, thereby improving the detection of weaker PMCs.
>
> Thus, we added some description to Section 4.4 (line 496 to 500) as:

*"…WFUI with only one camera cannot achieve the level of CIPS with four cameras when detecting small particles in weak PMCs. However, using a single camera significantly reduces costs and minimizes payload space. Also, WFUI is potential to be installed on multiple CubeSats to detect the same sampling PMCs region with variation scattering angle when different CubeSats are orbiting in the same orbit with a certain delay time interval. This would allow obtaining multiple views of each position from different SCA, thereby improving the detection of weaker PMCs. …"*

**3. Page 7, lines 171-173: Why not use an observed solar reference spectrum? One example is the TSIS-1 Hybrid Solar Reference Spectrum (HSRS), available at the LASP Solar Irradiance Datacenter (https://lasp.colorado.edu/lisird/).**

Thanks for the reviewer's good comment and instruction. We downloaded the daily average solar spectrum data from "https://lasp.colorado.edu/lisird/data/gsfc_composite_ssi" to recalculate the simulation result. This dataset covers the wavelength range of 120.5-499.5 nm in 1 nm bins, spanning from November 8, 1978, to July 24, 2022. It was generated by merging public irradiance data from nine satellite instruments: SME, Nimbus-7 SBUV, NOAA-9 SBUV/2, NOAA-11 SBUV/2, UARS SUSIM, UARS SOLSTICE, NOAA-16 SBUV/2, Aura OMI, and SORCE SOLSTICE. The daily variation of ultraviolet solar radiation from 2008 to 2012 is shown in the following figure.

[Figure]

After recalculation, we have updated Figures 6 to 9. Since the solar radiation calculated by using the Planck equation does not account for atmospheric absorption, the calculated value of solar radiation is higher than the actual reference spectrum. Therefore, the recalculated simulation signal has decreased after recalculation, as can be seen from Figures 6 and 7. However, this adjustment has a minimal impact on detection efficiency. The recalculated detection efficiency of the example orbit decreased from 81.64% to 81.09%. Figures 8 to 9 show slight changes, while the overall trend remains unchanged. The formula (19) for fitting the correlation curve between detection efficiency and IWC from 2008 to 2012 has been revised.

Thus, we have modified the sentence (lines 181 to 184) as:

*"… The WFUI was designed as a UV camera, so we utilized observed daily average solar UV spectral data to calculate solar radiation. This data can be acquired from the LASP Solar Irradiance Data Center (https://lasp.Colorado.edu/Liird/). The dataset covers the wavelength range of 120.5-499.5 nm in 1 nm bins, spanning from November 8, 1978, to July 24, 2022. It*

*was generated by merging public irradiance data from nine satellite instruments. …*"

We have modified the eq. 6 (line 187-188) as:

" $$S = \sum_{\lambda=\lambda_2}^{\lambda=\lambda_1}(E_\lambda \times t_\lambda) = \sum_{\lambda=\lambda_2}^{\lambda=\lambda_1}\left(E_\lambda \times \frac{t_{fmax}}{1+\left[\frac{2(\lambda-\lambda_0)}{FWHM}+\frac{\lambda}{FWHM}\frac{\theta_0^2}{n_e^2}\right]^2}\right) \qquad (6)$$ "

"… *where* $E_\lambda$ *is daily average solar spectra* …"

We have modified the sentence (line 391) as:

"… *the detection efficiency of PMCs by the WFUI was determined to be 81.09 %* …"

We have modified the eq. 19 (line 434) as:

" $\eta_d$= -2.211×e$^{\frac{IWC}{86.580}}$+1.299   (19)   "

The updated Figures 6 to 9 and the captions are modified as follows:

[Figure]

Figure 6: (a) Distribution of IWC with latitude and longitude for a single image detected by CIPS. (b) Distribution of the Photon number with latitude and longitude for a single image detected by WFUI.    (c) IWC of the orbit detected by CIPS. (d) Photon number corresponding to the single image pixels. (e) Gray value corresponding to the single image pixels. (f) The gray value of the orbit detected by WFUI. (g) Gray value of the single image after the denoising process. (h) Distribution of gray value with latitude and longitude for the single image after the denoising process. (i) Gray value of the orbit after the denoising process of WFUI.

[Figure]

Figure 7: (a) Distribution of IWC on August 3, 2011, detected by CIPS. (b) Simulating distribution of gray value.

[Figure]

Figure 8: (a) Average daily variation of IWC in PMCs season from 2008 to 2012. (b) Average daily variation of detection efficiency in PMCs season. (c) Variation of IWC with latitude in the northern hemisphere during the PMCs season from 2008 to 2012. (d) Variation of IWC with latitude in the southern hemisphere. (e) Variation of detection efficiency with latitude in the northern hemisphere. (f) Variation of detection efficiency with latitude in the southern hemisphere.

[Figure]

Figure 9: Correlation between the IWC and detection efficiency in the PMCs season for both hemispheres.

**4. Page 8, line 188: An optical thickness as large as 3 seems very high for a PMC, particularly in nadir viewing geometry. A recent paper by Lubken et al. [2024] (https://doi.org/10.1029/2023GL107334) gives a mean optical depth for PMCs of ~0.04-0.05 at 126 nm and 69 degrees N latitude for the year 2020.**

Thanks for the reviewer's suggestion. The optical thickness of PMCs has been adjusted to 0-0.3. We appreciate the reference you mentioned which is extremely helpful, and we have modified the sentences (lines 196-198) as:

"...*The optical thickness of PMCs is usually 0–0.3, which is consistent with the approximation of single cattering calculations. Generally, mean optical depth of PMCs from observation can be ~0.04-0.05, and the maximum value can reach 0.2-0.3 (Lubken et al., 2024)* …"

We have added this reference to the end:

"*Lubken, F. J., Baumgarten, G., Grygalashvyly, M., and Vellalassery, A.: Absorption of solar radiation by noctilucent clouds in a changing climate, Geophys. Res. Lett., 51(8), e2023GL107334, https://doi.org/10.1029/2023GL107334, 2024.*"

**5. Page 16, lines 371-373: This statement sounds like you will subtract a calculated noise term and assume that a positive residual signal represents a PMC detection. But the total observed signal also contains a background component determined by the amount of stratospheric ozone, which varies with time and latitude. Fluctuations in ozone could lead to false detections in your algorithm. Can you discuss how you would address this concern?**

Thanks for pointing this out. We have added several references and discussed this concern of the reviewer.

[revised manuscript text omitted]

**6. Page 18, lines 420-423: This analysis and the results shown in Figure 8 (panels (b), (e), (f)) suggest that WFUI would be less effective in observing the early and late portions of a typical PMC season, as well as latitudes equatorward of 65 degrees.**

Thanks for pointing this out. We appreciate for your analysis, and we have added this analysis into this paragraph (lines 438-440) as:

"*... From Fig. 9, there is a strong correlation between the IWC of the PMCs and the detection efficiency, with the numerical distribution following an exponential function pattern. When the IWC was relatively low, the detection efficiency increased rapidly as the IWC increased. However, once the IWC reached a higher level, the rate of increase in the detection efficiency decreased with further increments in the IWC. This analysis and the results shown in Figure 8 (panels (b), (e), (f)) suggest that WFUI would be less effective in observing the early and late portions of a typical PMCs season, as well as latitudes equatorward of 65 ° …*"

**7. Page 20, lines 454-458: These results indicate that changing instrument parameters doesn't improve the detection efficiency at the start or end of a PMC season. Did you evaluate these options at different latitudes?**

Thank you for pointing this out. We are also very interested in this issue. Consequently, we conducted a further parameter sensitivity analysis on detection efficiency at different latitudes during various periods of the PMCs season. We have also updated the title of section 4.4 to

"Parameter Sensitivity Analysis and Discussion". The results are shown in the figure below. We added some description to Section 4.4 (lines 483 to 491) as:

"… *The distribution of PMCs exhibits a strong latitude dependency. To further investigate the impact of parameter changes on detection efficiency at different latitudes, we conducted a sensitivity analysis of detection efficiency for PMCs across various latitudes and seasonal periods. Specifically, we divided the latitude range from 55 ° to 85 ° into 5 ° intervals and analyzed the detection efficiency of PMCs during various periods using the same orbit and parameters. The results are shown in Figure 11. Figure 11 shows that during the early and late stages of PMCs season, parameter changes have minimal impact on detection efficiency across different latitudes. During the early stage of the PMCs season, the appearance of PMCs is infrequent and faint, resulting in a detection efficiency of less than 10% during this period. However, during the early-mid, mid, and mid-late stages of the PMCs season, parameter changes significantly affect detection efficiency in the latitude range of 65 ° to 85 °, while the impact is relatively limited in the latitude range of 55 ° to 65 °. …*"

The added Figure 11 and the captions are shown as follows:

[Figure]

Figure 11: (a)-(e) Variation in detection efficiency with the FOV at different latitudes during different periods; (f)-(j) Variation in detection efficiency with the satellite altitude at different latitudes during different periods; (k)-(o) variation in detection efficiency with simultaneous changes in the FOV and satellite altitude at different latitudes during different periods.
* * *
Special thanks to the reviewer for his/her good comments.
* * *
We have tried our best to revise and improve the manuscript and made a few changes in the manuscript according to the reviewer's good comments. Again, many thanks for your valuable comments and suggestions. We would like to have our paper at your disposal.

We appreciate for reviewer's spending more time on reviewing over paper and offering valuable suggestions and hope that the modification and corrections will meet with approval and hoping that our paper will be published in *Atmospheric Measurement Techniques* as soon as possible.

We look forward to your information about my revised paper.

Yours sincerely,
Ke Ren and Haiyang Gao

---

## Author Comment (AC3)

Ke Ren and Haiyang Gao
School of Atmospheric Physics
Nanjing University of Information
Science and Technology
Nanjing, 210044, P. R. China
E-mail: renke0412@gmail.com

June 10th, 2024

Re:      Manuscript Number: AMT-2023-186
Title:    Simulation and detection efficiency analysis for polar mesospheric clouds measurements
            using a spaceborne wide field of view ultraviolet imager
Author: Ke Ren; Haiyang Gao; Shuqi Niu; Shaoyang Sun; Leilei Kou; Yanqing Xie; Liguo Zhang;
            and Lingbing Bu

Dear Reviewer,

Thank you for your review of the manuscript entitled "Simulation and detection efficiency analysis for polar mesospheric clouds measurements using a spaceborne wide field of view ultraviolet imager" pending revision as well as for your valuable suggestions to improve the paper. We have tried every means possible to improve the presentation and increased the readability. We have also made all the changes suggested by the reviewers and addressed all the comments in the notes below. Please note that the reviewer's comments are shown in bold type and our responses in plain type.

Best regard,
Ke Ren and Haiyang Gao

**This paper proposed a compact and cost-effective wide field-of-view ultraviolet imager (WFUI) for future PMCs observation and a forward model to evaluate the detection capability and efficiency of the WFUI. The lightweight imager mounted on the further small Cubesat for PMCs observations has great application prospects. The paper is well-written and comprehensive.**

**Comments:**

**1. Line 300: Fig.5 should be Fig.4.**

Thanks the reviewer for pointing this out.

We have modified the error in this sentence (line 314).

"*... The data were fitted, and the resulting image is shown in Fig. 4 ...*"

**2. Some discussions should be mentioned in the manuscript. The paper should look into the possible problems in the on-orbit calibration and geolocation of the observations, such as the exposure time for the CCD camera and the orbital altitude of the satellite.**

Thanks for pointing this out. We have discussed this issue, and the information has been incorporated.

For in orbit calibration, we have added some description to Section 2.6 (lines 349 to 354) as:

"*... On-orbit calibration primarily focuses on camera flat-fielding and normalization ... Ideally, obtaining an accurate estimate of the Δ-flat field requires a uniformly illuminated camera image. On-orbit is condition is best approximated at the subsolar point and in nadir viewing geometry (Lumpe et al., 2013). Images are captured multiple times throughout the year at the subsolar point with the camera for calibration of other images. For each measured subsolar image, a simulated image is calculated from a Rayleigh scattering forward model using identical viewing geometry. The measured image is then divided by the model image, and the resulting ratio is normalized to unity at the image center, to isolate the pixel-to-pixel variation...*"

We modified the title of Section 4.4 to "Parameter Sensitivity Analysis and Discussion", and have added some description to lines 457 to 461 as:

"*... The satellite velocity V varies with changes in the satellite altitude H, given by* $V = \sqrt{\frac{GM_e}{R_e + H}}$ , *where G is the gravitational constant, $M_e$ is the mass of the Earth, and $R_e$ is the radius of the Earth. As orbit altitude increases, satellite speed decreases, resulting in smaller phase shifts during image capture with the same exposure time, and reduced distances moved with the same measurement cadence. We can adjust the exposure time and measurement cadence to maintain the geolocation of the observations. In addition, while altering the satellite altitude, factors such as photon reception efficiency, and CCD pixel resolution change ...*"

For the identification of pixels with weak PMCs and multi CubeSats observations, we also added some description to Section 4.4 (lines 491 to 496) as:

"*... WFUI with only one camera cannot achieve the level of CIPS with four cameras when detecting small particles in weak PMCs. However, using a single camera significantly reduces costs and minimizes payload space. Also, WFUI is potential to be installed on multiple CubeSats to detect the same sampling PMCs region with variation scattering angle when different CubeSats are orbiting in the same orbit with a certain delay time interval. This would*

*allow obtaining multiple views of each position from different SCA, thereby improving the detection of weaker PMCs. ...*"
* * *
Special thanks to the reviewer for his/her good comments.
* * *
We have tried our best to revise and improve the manuscript and made a few changes in the manuscript according to the reviewer's good comments. Again, many thanks for your valuable comments and suggestions. We would like to have our paper at your disposal.

We appreciate for reviewer's spending more time on reviewing over paper and offering valuable suggestions, and hope that the modification and corrections will meet with approval and hoping that our paper will be published in *Atmospheric Measurement Techniques* as soon as possible.

We look forward to your information about my revised paper.

Yours sincerely,
Ke Ren and Haiyang Gao